# Novel Chemical and Biological Insights of Inositol Derivatives in Mediterranean Plants

**DOI:** 10.3390/molecules27051525

**Published:** 2022-02-24

**Authors:** Laura Siracusa, Edoardo Napoli, Giuseppe Ruberto

**Affiliations:** Istituto di Chimica Biomolecolare, Consiglio Nazionale delle Ricerche, Via Paolo Gaifami 18, 95126 Catania, Italy; laura.siracusa@icb.cnr.it (L.S.); edoardo.napoli@icb.cnr.it (E.N.)

**Keywords:** inositols, natural occurrence, biological role, bioactivity

## Abstract

Inositols (Ins) are natural compounds largely widespread in plants and animals. Bio-sinthetically they derive from sugars, possessing a molecular structure very similar to the simple sugars, and this aspect concurs to define them as primary metabolites, even though it is much more correct to place them at the boundary between primary and secondary metabolites. This dichotomy is well represented by the fact that as primary metabolites they are essential cellular components in the form of phospholipid derivatives, while as secondary metabolites they are involved in a plethora of signaling pathways playing an important role in the surviving of living organisms. *myo*-Inositol is the most important and widespread compound of this family, it derives directly from d-glucose, and all known inositols, including stereoisomers and derivatives, are the results of metabolic processes on this unique molecule. In this review, we report the new insights of these compounds and their derivatives concerning their occurrence in Nature with a particular emphasis on the plant of the Mediterranean area, as well as the new developments about their biological effectiveness.

## 1. Introduction

### 1.1. Inositols

Inositols (Ins) constitute a particular and peculiar class of natural metabolites [1]. From a chemical point of view, they can defined polyols or cyclitols being the basic structure a cyclohexane with an hydroxy group bound to each carbon atom of the hexanic ring, therefore they can be called 1,2,3,4,5,6-cyclohexanehexols [2,3]. In this respect, these compounds have a structure very similar to the cyclic form of monosaccharides like glucose, and accordingly they are also defined sugar alcohols [4].

This particular typology of structure with six chiral centers assumes the presence of 64 potential stereoisomers, however, owing to symmetry reasons the real isomers are nine as reported in Figure 1. *Myo*-, d-*chiro*-, l-*chiro*-, *muco*-, *scyllo*-, and *neo*- (**1**–**6**) are the isomers naturally occurring, whereas *allo*-, *cis*-, and *epi*- (**7**–**9**) are synthetic compounds [5]. *Myo*-inositol (*myo*-Ins) is the most common component, followed by *chiro*-inositol (*chiro*-Ins) [4,6,7].

Inositols are widespread in all eukaryotes, being involved in a large number of biological processes [8]; also for this reason they have been enclosed in the group of vitamins B [9]. However, unlike the well known vitamins, which can not be synthetized by mammals, inositols are produced in the liver and kidney of mammals at the rate of about 4 g/day, therefore it not would be necessary to assume them by diet [9].

Further inositol derivatives are represented by inositol methyl ethers, bornesitol (**10**), ononitol (**11**), sequoytol (**12**) are monomethyl ethers of *myo*-inositol [10], while viscumitol (**13**) and dombonitol (**14**) are dimethyl ethers of *myo*-inositol [11,12]. d-Pinitol (**15**), quebrachitol (**16**) and pinpollitol (**17**) are methyl- and dimethyl-ethers of d-chiro-inositol [10,13], brahol (**18**) is a monomethyl ether of *allo*-inositol [10,14]. Figure 2 reports the structure of the aforesaid methyl ether compounds. Other derivatives are represented by inositol glycosides as ciceritol (**19**), a pinitol digalactoside [15] (Quemener and Brillouet, 1983) and fagopyritols (**20**), a group of galactosyl derivatives of d-chiro-inositol [16,17,18]. Figure 2 reports the structures of these last components.

Inositols (Ins) are not present only as free components but are also incorporated in several biomolecules. A first important class of these derivatives is represented by inositol pyrophosphates (PP-InsPs) determined for the first time in 1914 and characterized by the presence of a pyrophosphate moiety on the hydroxyl groups of the inositol ring [19,20,21]. These substances are involved in a large number of cellular functions of eukaryotes [22,23]. The inositol hexakisphosphate (InsP6) well known as phytic acid (**21**) (Figure 3), is the main phosphate storage compound in plants [24,25,26]. This compound shows a great affinity to several minerals, such as iron, zinc and calcium, reducing their intestinal absorbtion, and for this reason phytic acid and its salts are also considered anti-nutritional dietary components [25,26]. Several inositol phosphates have been isolated and characterized mainly from seeds, cereals and legumes, being present as mixed salts (calcium-magnesium-potassium), named also phytins [21,27,28].

Another important class of biomolecules involving inositols is represented by sphingolipids, one of the most significant lipid class present in the plant plasma membrane [29]. These compounds, as the inositols, are ubiquitous to eukaryotes and assume a strategic importance in living organisms being involved in growth, cellular signaling, stress responses etc. In plants and fungi, the main sphingolipds are glucosyl inositol phosphoryl ceramides (GIPCs) [30,31,32,33], which are not present in mammalian organisms [34] (Figure 4).

A further class of biomolecules containing inositol moiety is represented by the so called archedityl inositols, a class of polar lipids isolated from extremophile Archean [35,36]. These components structurally similar to the previous sphingolipid (Figure 4) are also known as archaeal diether lipids showing a possible role as drugs and possessing particular structural features.

### 1.2. The Dual Nature of Primary and Secondary Metabolites of Inositols

As mentioned earlier, inositols can be found not only as widespread compounds within the vegetable kingdom but also in mammals, including human beings [5], this contributes in defining them as primary metabolites, just like sugars from which they derive. The numerous studies on inositol metabolism converge around the biosynthesis of *myo*-inositol, the most omnipresent of these compounds, its subsequent epimerization to other inositols, its methylation to give many *O*-methyl esters, incorporation into cell walls, and so on [37]. In addition, the discovery that *myo*-inositol plays a pivotal role in growth and development of plants, microorganisms and certain yeasts further prompted renewed interest in its biochemical and biological features. Nowadays, it is known that *myo*-inositol derives directly from d-glucose (in the form of d-glucose-6-phosphate); briefly, this conversion involves the cyclization of d-glucose-6-phosphate to inositol 3 phosphate, the loss of phosphate and the final release of free *myo*-inositol (Figure 5). This two-step pathway is the only recognized for *myo*-inositol biosynthesis in algae, fungi, cyanobacteria, higher plants and animals. All known inositols, including stereoisomers and derivatives are the results of metabolic processes on this unique molecule [38].

Finding relevant, sustainable and possibly cheap sources of inositols is a current challenge in phytochemistry, due to their biological significance, as well as a crucial step that allows for plant selection for a convenient industrial exploitation. As it has been already discussed, the high number of stereogenic centers in the inositol scaffold gives rise to a number of possible stereoisomers; six of them (*myo*-, *scyllo*-, *muco*-, *neo*, d-*chiro*- and l-*chiro* inositol) are naturally occurring compounds, while the other three (*allo*-, *cis*- and *epi*-inositol) are derived directly from *myo*-inositol [39]. When considering the methyl ether derivatives of inositols, they are rightly regarded as plant secondary metabolites, that is, compounds that are not directly involved in the normal growth but which play an important role in the defense against unfavorable environmental conditions. As secondary metabolites and in opposition to the ubiquitous nature of *myo*-inositol, they are produced and accumulated only in certain species, generally higher plants. Examples of this category of inositols are the mono- and di- *O*-methylated inositol derivatives bornesitol (**10**), ononitol (**11**), sequoyitol (**12**), and pinpollitol (**17**) [40].

Main aim of this review is to give an updated summary of the studies carried out in the last five years on the inositols. Particular (or special) attention has been paid to the inositol derivatives isolated from Mediterranean plants and to the screening of novel biological activities evaluated in the considered period.

## 2. Natural Occurrence of Inositols

### 2.1. Natural Sources of Inositols: Some Examples

In a recent work, Ratiu et al. [41] analysed 52 plant sources coming from 40 different species in search for cyclitols, thus finding 37 new sources of these compounds. In their study, medicinal plants and herbs were analyzed (*Taraxacum officinale*, *Laurus nobilis*, *Sambucus nigra*, *Salvia officinalis*, *Chamomilla recutita*, *Hypericum perforatum*, *Mentha piperita*), as well as spices (*Curcuma longa*, *Trigonella foenum-graecum*, *Zingiber officinale*, *Capsicuum annuum*) and table vegetables (*Daucus carota*, *Lactuca sativa*, *Brassica oleracea*, *Solanum tuberosum*). As expected, all investigated plants contained *myo*-inositol in variable amounts; in particular, cinnamon, lettuce and blueberry fruits were found to contain quantities of this compound (1.21, 1.07 and 0.96 mg/g dry vegetable material, respectively). *Allo*-inositol, one of the four possible stereoisomers directly derived from *myo*-inositol, was found in 14 samples over 52 with the richest sources being blueberries (10.84 mg/g dry vegetable material). Other inositols reported in this study were the already mentioned ononitol (blueberries, wild garlic, garlic, kale, mint) bornesitol (goldenrod flowers) and *scyllo*-inositol, which was detected in 30 over 52 samples and whose highest content was found in carrot (0.81 mg/g dry vegetable material). A similar study was carried out on 17 edible vegetables belonging to the families of Asteraceae (chicory, endive, escarole, artichoke, iceberg lettuce, oak leaf lettuce, lollo rosso lettuce, romaine lettuce, cresta lettuce, lactuca batavian lettuce), Amarantaceae (spinach and beet root), Amarylidaceae (onion), Brassicaceae (radish and cabbage), Dioscoreaceae (purple jam), and Solanaceae (eggplant) [42]. d-*Chiro*-inositol was identified and quantified in all members of the Asteraceae family with quantities ranging from 3.1 to 32.6 mg/100 g dry product, whilst *scyllo*-inositol was detected in noteworthy amounts only in purple yam (28.3 mg/100 g dry product) and chicory leaves (5.3 mg/100 g dry product). The Fabaceae (or Leguminosae) family seems to be particularly rich in inositols: free inositols (*myo*- and *chiro*-) and methyl-inositols have been detected in edible legume seeds, together with galactosyl-inositols (galactinol isomers, galactopinitols, galactosyl-ononitol, fagopyritols). As example, Ruiz-Aceituno and co-workers [43] analysed some of the most common members of the Fabaceae family (black-eyed peas, buckwheat, carob pods, chickpeas, grass peas, lentils and soy beans); their study revealed again *myo*-inositol in all the legume extracts analysed, particularly in chickpeas in considerable amounts (1.22 mg/g sample). d-*Chiro*-inositol was found at much lower concentrations in almost all the matrices investigated, excluding black-eyed and grass peas. Apart from carob pods, which are considered as the main natural source of d-pinitol with more than 100 mg/g sample, the authors found this metabolite also in soybeans, chickpeas, and lentils. Another methyl-inositol, identified as ononitol, was identified in black-eyed peas (2.03 mg/g sample).

A microwave-assisted extraction (MAE) was carried out by Zuluaga et al. [44] in order to obtain inositols-rich extracts from legume byproducts coming from edible legume industrial processing in the view of their possible valorization in alternative sectors than animal feed. To this end, the authors analysed fifteen different samples including pods and seeds from ten leguminous plants; the results they reported showed that *myo*-inositol was detected in all pods (2.03 mg/g sample), d-*chiro*-inositol was identified only in basul pod extract and pinitol in soybean pod extract in remarkable amounts (34.52 mg/g dry sample). The authors conclude that soybean pods are worth of further investigations as promising source of d-pinitol. Another source of inositols strictly related to the edible members of the Fabaceae family is that of bean sprouts, that is, the product of bean germination. According to several studies, germination has been proposed as a simple natural method to improve the nutrient composition and certain functional properties of the corresponding seeds [45].

Da Silva Ribeiro and coauthors [46] studied the nutritional properties, including the presence of inositols in two different bean sprouts, *Vigna unguiculata* (cowpea) and *Phaseolus vulgaris* (common bean). The study was carried out by following a time course of *myo*-inositol, d-pinitol and ononitol levels in the species under investigation during sprouting. Their results showed an increase in *myo*-inositol levels in both species when compared to quiescent seeds; on the contrary, d-pinitol was identified only in seeds while disappeared during sprouting. Another member of the Fabaceae family having attracted interest for its unique features, including the presence of several inositols, is alfa-alfa (*Medicago sativa*), commonly used from the ancient times as fodder crop as well as a traditional remedy in the treatment of various illnesses. In a reasonably recent study, Al-Suod and coworkers [47] investigated the presence and content of four biologically relevant inositols (*myo*-inositol, d-*chiro*-inositol, *scyllo*-inositol and d-pinitol) in different morphological parts of alfa-alfa (leaves, stems, flowers and roots) plant using targeted extraction methods. The results they obtained showed a preferred accumulation of d-pinitol, over the other inositols analysed, in all the morphological parts under study; alfa-alfa leaves were also found the only part of the plant containing *scyllo*-inositol. Under a quantitative point of view, roots of *Medicago* were identified as the richest source of inositols of the whole plant. As discussed earlier in this section, inositols are widespread within the plant kingdom. Some apparently unusual source of these compounds include chamomile (Asteraceae) flowers (d-pinitol, *chiro*-inositol, *neo*-inositol, d-(−)-bornesitol, ononitol, *scyllo*-inositol, *myo*-inositol) [48]; needles belonging to five different conifer genera (*Abies*, *Larix*, *Picea*, *Pinus* (Pinaceae) and *Juniperus* (Cupressaceae) (*myo*-inositol, d-pinitol, sequoyitol) [49]; leaves from the exotic medicinal plant *Hancornia speciosa* Gomes (Apocynaceae) (bornesitol) [50]; blue tansy (*Phacelia tanacetifolia* Benth. commonly known as lacy phacelia from the Boraginaceae family) (*allo*-inositol, *scyllo*-inositol, *myo*-inositol) [51]; licorice (*Glycirrhiza glabra*, Fabaceae) leaves [52,53].

Table 1 summarizes the occurrence of inositols in plants, listed according to their belonging to different families.

A special mention as source of inositols deserves a plant-animal hybrid product, that is, bee honey, where quercitol (1,3,4/2,5-cyclohexanepentol), d-pinitol, 1-*O*-methyl-*muco*-inositol and *muco*-inositol have been identified for the first time only in 2004 by Sanz et al. [57]. In their work, 28 different honeys were analyzed finding that these compounds were present only in a limited number of samples, likely those collected by bees from wild plants. They therefore concluded that these molecules could be used as markers for honey identification and traceability. More recently, Ratiu et al. [58] have resumed and perfected the use of inositols as markers in honey traceability; they concluded that, even if present in the matrices as minor components, these compounds can be effectively used to discriminate honeys from different origins.

### 2.2. Accumulation of Inositols in Response to Biotic and Abiotic Stresses

Two distinctive traits of secondary metabolites are their variegate distribution in plants depending on several factors (genetics, environment, climate) and their response to biotic and abiotic stresses, both natural or induced [59]. In line with their nature, inositols accumulation may vary under unfavorable conditions, particularly drought stress, as a number of studies indicate that cyclitols are potentially important osmolytes in plants. Streeter and coworkers [54] reported a significant increase in d-pinitol levels in soybean (*Glycine max*) associated with adaptation to dry areas of China; similarly, Liu and Grieve [56] studied the effect of salt stress on two statice cultivars, *Limonium perezii* cv. Blue Seas and *L. sinuatum* cv. American Beauty. Their analyses revealed, for the first time in these species, the presence of *chiro*-inositol in addition to *myo*-inositol; furthermore, significant variations in *chiro*-inositol level as salinity increased suggested that this compound contributes to osmotic adjustment in the stressed plants, and that *chiro*-inositol response might be a physiological process for *Limonium* salt adaptation. Finally, in a very recent study, Foti et al. [55] investigated the metabolic response of lentil (*Lens culinaris* Medik, Fabaceae) when subjected to osmotic drought stress induced by PEG (polyethylene glycol). The authors found that not only a number of compounds, including *myo*-inositol, undergo differential accumulation, but also that the adaptive metabolic responses to osmotic drought stress operate under strong genetic control.

## 3. Biological Activities of Inositols

Clinicians’ interest in the biological activities of inositols has very ancient roots, although in the last thirty years the crucial role played by these important biomolecules has begun to be understood [60].

First of all, as part of the membrane in the form of phospholipid derivatives they are essential components of the cells themselves. As second messengers, they play a very important role in several transduction pathways such as metabolic modulation, endocrine modulation, cell growth and signal transduction downstream neurotransmitter stimulation [61]. Among the nine existing stereoisomers, d-*chiro* and *myo*-inositol are the two main present in human body [62]. About 1 g/day of *myo*-inositol is provided by normal and balanced diet, but this quantity, which is not sufficient for the daily needs, is compensated by the endogenous production (about 4 g/day), especially by the kidneys. Endogenous *myo*-ins production starts with the isomerization of glucose-6-phospate as reported in Figure 5. Thanking tissue-specific epimerase enzymes, *myo*-ins can be converted into its stereoisomer d-*chiro*-inositol under adequate molecular stimulation (insulin), in order to obtain the proper balance between the two [60,63]. To the maintenance of the right qualitative-quantitative balance of the two main inositols therefore contributes to multiple factors including food-dependent absorption, biosynthesis and cellular uptake. The depletion of cellular *myo*-ins levels caused by imbalances of the afore-mentioned factors has been related to several chronic diseases, including metabolic syndrome, diabetes, some type of cancer and polycystic ovary syndrome (PCOS) [64]. It is now clear the importance of maintaining the physiological level of inositols in our body through the correct functioning of endogenous production mechanisms and through a balanced diet. Inositols are successfully applied in treatment of polycystic ovarian syndrome and insulin resistence and obesity as well as showing anti-atherogenic, anti-oxidative, anti-inflammatory and anti-cancer properties [10].

### 3.1. Inositols and Polycystic Ovary Syndrome (PCOS)

Polycystic ovary syndrome (PCOS) is the most common endocrine disorder in women. It typically presents with symptoms of menstrual disturbances, hirsutism, anovulation and consequent anovulatory infertility and recently is recognized to be a major risk factor for the development of diabetes mellitus. PCOS is associated with metabolic disorders such as hyperinsulinaemia, insulin resistance and dyslipidemia [65]. The PCOS tissues are not able to synthesize inositols and this may contribute to insulin resistance and hyperinsulinaemia. The latter can be explained by a malfunction of the epimerase responsible for the conversion of *myo*-ins into d-*chiro*-ins. From a series of experimental evidences, it is clear that an imbalance in the ratio between these two metabolites mediated by an altered epimerase activity plays a key role in the development of insulin resistance [62]. *Myo*-ins alone or in combination with d-*chiro*-ins showed to exert an effect against PCOS syndrome improving metabolic and ovarian function [66], especially in the 40:1 ratio exactly what we can find in the plasma [67,68] (Figure 6). However, the most recent position seems to favor the fact that the choice of the respective concentrations of the two isomers must be evaluated with caution [69], evaluating patients’ anamnesis taking into account familial diabetes, overweight and/or obesity [70].

### 3.2. Inositols in Metabolic Syndrome and Diabetes

Diabetes is a metabolic disorder characterized by chronic hyperglycemia. Together with obesity, hypercholesterolemia and high blood pressure are considered the most dangerous risk factors than often lead to heart attacks. The simultaneous combination of two or more of these risk factors gives rise to the so-called metabolic syndrome.

Inositols, with particular reference to *myo*-inositol, d-*chiro*-inositol and d-pinitol, have been show to possess health promoting properties such us improving lipid profiles [71] and insulin-like effects and for these reasons may have a positive effect on patients with metabolic disorders correlatable with metabolic syndrome [72,73]. In addition, the results of some randomized controlled studies indicate the beneficial effect of inositol supplementation on the improvement of systolic and diastolic blood pressure especially in patients with metabolic syndrome [74].

The effect of supplementation of combined *myo*-inositol hexakisphosphate and *myo*-inositol to the liver of streptozotocin-induced type 2 diabetic rats has been conducted by Foster and collaborators [75] showing that the combined supplementation normalized liver lipid status and consequently the combination of the two inositols may be considered as a potential agent for the effective management of type 2 diabetes and associated metabolic disorders. A pilot study involving sample of patients with type 2 diabetes mellitus (T2DM) with suboptimal glycemic control already treated with glucose-lowering agents and for three months with a combination of *myo*-inositol (550 mg) and d-*chiro*-inositol (13.8 mg) orally twice a day as add-on supplement has been conducted by Pintaudi and collaborators [76]. Results of this study show after three months of treatment fasting blood glucose and HbA1c levels significantly decreased compared to control. A more recent systematic review and meta-analysis of 20 randomized controlled trials carried out of summarize the effects of inositol on glucose homeostasis in different clinical conditions show that this kind of supplementation decreases blood glucose through an improvement in insulin sensitivity that is independent of weight [77]. These evidences suggest that *myo*-inositol and d-*chiro*-inositol supplementation promotes beneficial effect on glycemic parameters of patients at risk or with T2DM [62].

*Myo*-inositol treatment in early pregnancy is associated with a reduction in the rate of gestational diabetes mellitus (GDM) and in the risk of preterm birth and macrosomia as reported in a secondary analysis of databases from three randomized, controlled trials with 595 women at risk for gestational diabetes mellitus enrolled. Despite the authors highlight some weaknesses in the design of primary trials base of the study, the results confirming a significant reduction of gestational diabetes mellitus rate in women who received *myo*-inositol in comparison with placebo, demonstrating also a reduction of preterm birth rate and in the rates of macrosomia [78]. *Myo*-inositol administered orally as an adjuvant to a modified diet has been shown to be effective in controlling glycemia even in a study of 100 women with gestational diabetes mellitus. The early diagnosis of the disease and the administration of *myo*-inositol at a dose of 1 g twice a day allowed a decrease in the need for additional pharmacological therapies compared to the control group [79]. Previous results are also confirmed by a systematic review and meta analysis on four randomized controlled trials with 586 patients included conducted by Guo and collaborators concluding that *myo*-inositol is related to lower incidence of gestational diabetes mellitus in pregnant women with high risk of this condition [80]. Co-administration of *myo*-inositol and α-lactalbumin has been evaluated to improve insulin resistance and birth outcome in women with gestational diabetes in a randomized controlled study enrolling 120 patients at the dose of 2 g and 50 mg twice a day respectively. This combination of *myo*-inositol and α-lactalbumin reduce insulin resistance and excessive fetal growth in women with gestational diabetes mellitus and a reduced rate of insulin treatment and pre-term birth was demonstrated in the treated group [81]. As declared by Authors, the mode of action of myo-inositol is still unclear but probably myo-inositol supplementation may increase insulin sensitivity by making more available a second messenger of insulin (phosphatidylinositol) improving glucose uptake from the bloodstream to the cells reducing contextually the release of free fatty acid from adipose tissue. A study with different purposes confirmed the positive correlation between the administration of α-lactalbumin and the absorption of *myo*-inositol through the interaction with the gut microbiota [82]. GDM is associated with a pro-inflammatory state and increased oxidative stress, which are both involved in vascular damage in diabetes. In an in vitro model of human umbilical vein endothelial cells (HUVECs), *myo*-inositol anti-inflammatory and antioxidant potential effects have been evaluated [83]. The authors conducted their study using HUVECs obtained from GDM women treated with diet plus *myo*-inositol supplementation during pregnancy as compared to HUVECs obtained from GDM women treated with diet only. Based on their findings the authors report that *myo*-inositol supplemented in vivo, significantly reduced levels of monocyte cell adhesion, adhesion molecule exposure, and intracellular reactive oxygen species levels in the basal state as compared to those obtained from women treated by diet-only.

### 3.3. Inositols in Neurodegenerative and Neurologic Disorders

In a recent review published by Surguchov [84] highlighted several epidemiological studies showing that patients with T2D have higher incidence of neurodegenerative disorders, pointing to the existence of a common mechanism for both. It seems that the dysregulation of the metabolic network leads to an age-related elevated risk of suffering of insulin resistance-related pathologies such as other metabolic disorders. As known insulin regulates a series of cognitive processes, such as memory formation, through its effects on glial–neuronal metabolic coupling. This is probably the reason why there is some case history of central insulin resistance is observed in neurological disorders, including Alzheimer’s disease (AD) and Down’s syndrome (DS) [85]. It was hypothesized that *myo*-inositol and *scyllo*-inositol are promising therapeutic agent for Alzheimer’s disease, because of its inhibitory effect on amyloid β protein (Aβ) aggregation and reduction of cerebral Aβ pathology [86,87]. A double-blind, placebo controlled, phase 2 study on the safety and pharmacokinetics of *scyllo*-inositol administered orally to young adults with DS shows a well tolerability and no safety findings although further studies are needed to prove its therapeutic efficacy as AD treatment in patients with DS [88].

Potential use of inositols for treatment of psychological symptoms as anxiety, depression and sleep disorders has been also investigated. The use of inositols on treatment of psychological symptoms on women affected by PCOS syndrome was critically reviewed by Cantelmi and collaborators [89]. Authors conclude that inositol already under study for the treatment of PCOS syndrome proved safe and really effective in depression, and so its contextual use in psychological issues related to PCOS could represent a novel and interesting approach, able to restore normal physiology and helping in alleviate psychological symptoms. Correlation between *myo*-inositol brain levels and anxiety/depression symptoms in young adult has been recently evaluated in two different in vivo studies by means of magnetic resonance. Results of the first study conducted on nineteen not medicated adolescent 9 of which with depression symptoms and 10 as control showed concentrations of *myo*-inositol lower frontal cortical regions among the depressed adolescents than in controls [90]. The second study conducted on 149 adolescents explored the mediating action of *myo*-inositol on the effects of traffic related air pollution (TRAP) on anxiety symptoms. Results show that high levels of TRAP exposure are associated with increased *myo*-inositol in brain, which in turn is associated with a greater degree of anxiety symptoms [91]. According to the results published by Shirayama et al. [92] there is a significant reduction of *myo*-inositol in the medial prefrontal cortex, hippocampus, and amygdala of patients with depression compared with normal control subjects in a study conducted on 22 drug naïve patients. The anxiolytic and anticonvulsant activity of d-pinitol has been assessed in an in vivo murine model showing a dose-dependent anticonvulsant effect and delayed the onset of convulsion. In this model, anticonvulsant effects of d-pinitol were evaluated with the pentylenetetrazole (PTZ)-induced convulsion test while ansiolytic affect by means of exploratory rearing test. Authors suggest the possible participation of the GABAergic system in the anxiolytic-like and anticonvulsant actions of d-pinitol [93]. The probable role of endogenous anticonvulsant agent in the central nervous system of *myo*-inositol has recently been speculated by Gamkrelidze and collaborators in a study that highlights the effect of local injection of *myo*-inositol on the epileptic after discharges induced by local electrical stimulation in the murine hippocampus [94].

### 3.4. Inositols and Sars-Cov-2

Nowadays Sars-CoV-2 became a primary healthcare emergency worldwide and for this reason the search for new treatments to reduce the severity of the prognosis of the most serious patients has become a priority. As known the most serious cases develop an acute respiratory distress syndrome (ARDS) characterized by high levels of pro-inflammatory cytokines especially IL-6. Starting from these assumptions, the authors recently formulated the hypothesis that *myo*-inositol is able to reduce the IL-6 dependent inflammatory response in patients who manifest ARDS [95]. According to the authors, the action of *myo*-inositol could be expressed in two ways: the first by stimulating the production of pulmonary surfactant and the second thanks to the ability of this molecule to down-regulate IL-6 levels [96,97]. Although promising, the data relating to this type of therapeutic approach are still very limited and require further investigations.

### 3.5. Inositols and Cancer

The protective and anticancer effects of inositols and in particular of *myo* inositol and its derivative hexaphosphate have been studied for many decades. Many of these were published in the period prior to the one considered in this review and for this reason they will not be taken into consideration. An overview of these studies has been excellently summarized by Wisniewsky and collaborators [98]. Studies with particular reference to colon cancer and colitis-induced cancer were collected and discussed by Vucenik and collaborators [99] and Weinberg and collaborators [100], respectively.

The anticancer activity of inositols has been defined as ‘broad spectrum’ thanks to their ability to modulate the whole cell cycle (progression, apoptosis, and differentiation) [101]. Furthermore, inositols seems able to play an important role improving the cure and decreasing side effects of cancer treatments [98]. *Myo*-inositol-trispyrophosphate is a non-cytotoxic molecule with tumour-reducing effects ascribed to its capability to promote vessel normalization, increases in oxygen partial pressure (pO_2_) which counteracts hypoxia inside the tumour and inverts its effects. The work of El Hafni-Rahbi and collaborators carried out on two tumor models (melanoma and breast cancer) shows that normalization induced by *myo*-inositol-tripyrophospate cause changes in the tumour microenvironment due to hypoxia compensation [102]. Due its action in tumor oxygenation and vascular protection, the use of *myo*-inositol-tripyrophosphate as adjuvant to increase the efficacy of ionizing radiation for successful radiation therapy has recently supported [103].

The studies conducted up to now on the effect of *myo*-inositol administration in animal trials lung cancer model have shown its positive impact through the downregulation of IL-6 levels which is believed to be involved in tumor progression. Human trials, instead, do not support *myo*-inositol alone as chemopreventive agent against lung cancer. In an in vivo study on transgenic mice, the effect on tumor progression of *myo*-inositol as supplement in the diet has been evaluated by Unver and collaborators [104] confirming *myo*-inositol as potent chemopreventive effect in a model of lung premalignancy and early cancer.

Breast cancer is one of the most common malignancy worldwide. New chemotherapy therapies and an increasing of early screening have made it possible to significantly reduce the mortality. In cancer patients, quality-of-life and adverse effects represent a very critical aspect during cancer journey and new therapeutic approaches in this direction are increasingly indispensable. Effects on local symptoms and quality of life-related symptoms of oral and topical administration of *myo*-inositol and inositol hexaphosphate respectively in patients undergoing surgery for breast cancer and eligible to adjuvant chemotherapy have been evaluated by Amabile and collaborators [105]. In their cohort of breast cancer patients, this combined treatment was able to improve local symptoms and quality-of-life related symptoms, which represent clinically relevant aspects associated with patient’s prognosis. The effectiveness of topical inositol heaphosphate treatments in mitigating chemotherapy-induced side effects and improving patients with ductual breast cancer quality of life has been evaluated in a double-blind randomized controlled trial [106]. *Myo*-inositol (200 mg/die) in combination with boswellic acid and betaine has been evaluated for the management of benign breast disorders in a pilot study recruiting 76 women. The preliminary results have shown that women suffering from mastalgia experience a significant clinical benefit when treated with proposed drug formula [107]. The effectiveness of inositol hexaphosphate as adjuvant in cancer therapy has been investigated on medulloblastoma cells models. Results show that inositol hexaphosphate treatment acts synergistically with cisplatin in medullo-blastoma cells both in vitro and in an in vivo xenograft model and suggest that administration of this molecule could be implemented as adjuvant therapy in G4 medullo-blastoma patients that are currently treated with cisplatin [108].

Some studies have correlated metabolic changes to the metastatic potential of cancer cells. In particular, results on osteosarcoma cells suggest the hypothesis that their metastatic behaviour is in part the result of metabolic alterations. To support this hypothesis, the effects of inositol pathway dysregulation, through the exposure of metastatic osteosarcoma cells to inositol hexaphosphate, were investigated [109]. Minimal effects on cell proliferation were registered, but authors observed reduced cellular glycolysis, down-regulation of PI3K/Akt signaling and suppression of osteosarcoma metastatic progression.

Colorectal cancer is one of the most commonly diagnosed cancers and the third leading cause of cancer mortality in the Western countries [110]. Inositol hexaphosphate has a suppressive effect on Caco-2 cells regulator of proliferation and apoptosis [111]. According to the authors, this result is obtained through the inhibition of the AKT/mTOR pathway. The protective effects of inositol hexaphosphate against 1,2-dimethyl hydrazine dihydrochloride-induced colon cancer in animal models has been also demonstrated by Yu and collaborators [112]. With reference to the results of the last ten years of studies and clinical practice inositol hexaphosphates and inositol supplementation can be a new option both for cancer prevention and cancer treatment [99]. Inflammatory bowel disease significantly increase risk for colorectal cancer and several studies indicates that inositol displays a strong inhibitory effect on carcinogenesis in patients with this syndrome [100]. In their overview, Weinberg and collaborators highlight how critical a nutritional deficiency of inositols can be for human health and emphasize the need for more data to establish minimum basal levels for these molecules to play their preventive role for many chronic diseases and for many types of cancer.

## 4. Concluding Remarks

The inositols beyond their classification as primary and/or secondary metabolites play a very important role in the lives of living organisms. They are, in fact, quite widespread in all eukaryotes, being involved in a large number of biological processes, probably also owing to their large presence in many biological pathways make these components particularly effective in many biological activities as reported in this review.

A number of works are present in literature on the occurrence of inositols in edible and non-edible vegetable matrices including herbs, spices and table vegetables. The majority of inositols, all derived from *myo*-inositol, possess the peculiar characteristics of secondary metabolites, and therefore can be used as chemotaxonomic and traceability markers, as hereby discussed.

The involvement of inositols in the modulation of many inflammatory processes and the insulin-like behavior of some of them makes them particularly interesting molecules from a pharmacological point of view. This is demonstrated by the number of clinical studies and the use of inositols in food supplementation in PCOS and metabolic syndrome patients. In spite of the numerous in vitro and in-vivo studies in animal models on the chemopreventive activity of inositols, their use in anticancer therapies as adjuncts still needs to be consolidated by more robust clinical data.

## Figures and Tables

**Figure 1 molecules-27-01525-f001:**
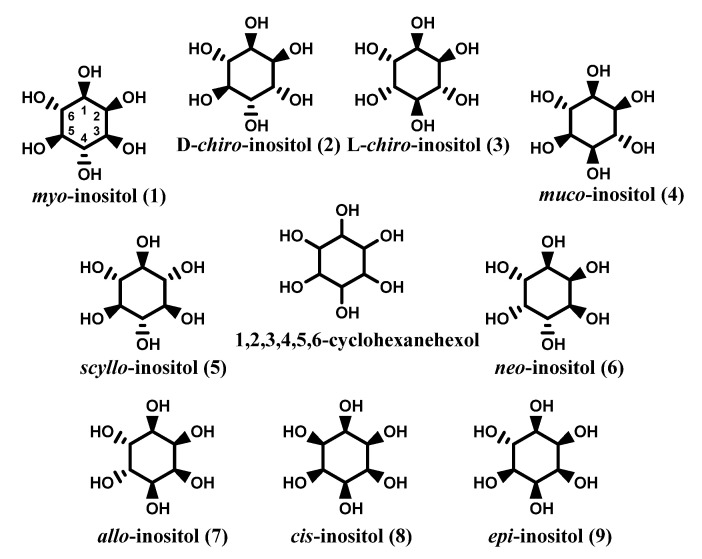
Inositols.

**Figure 2 molecules-27-01525-f002:**
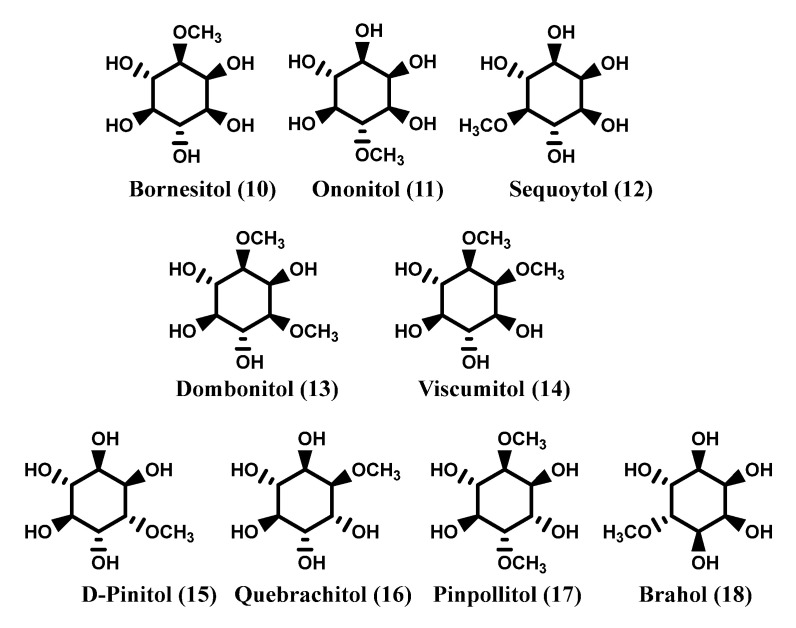
Methyl ethers derivatives of inositols.

**Figure 3 molecules-27-01525-f003:**
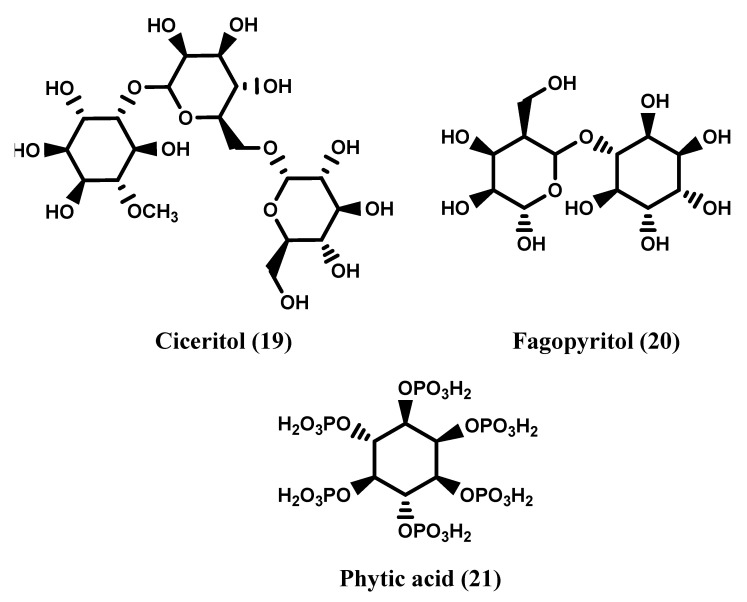
Glucosides derivatives of inositols and phytic acid.

**Figure 4 molecules-27-01525-f004:**
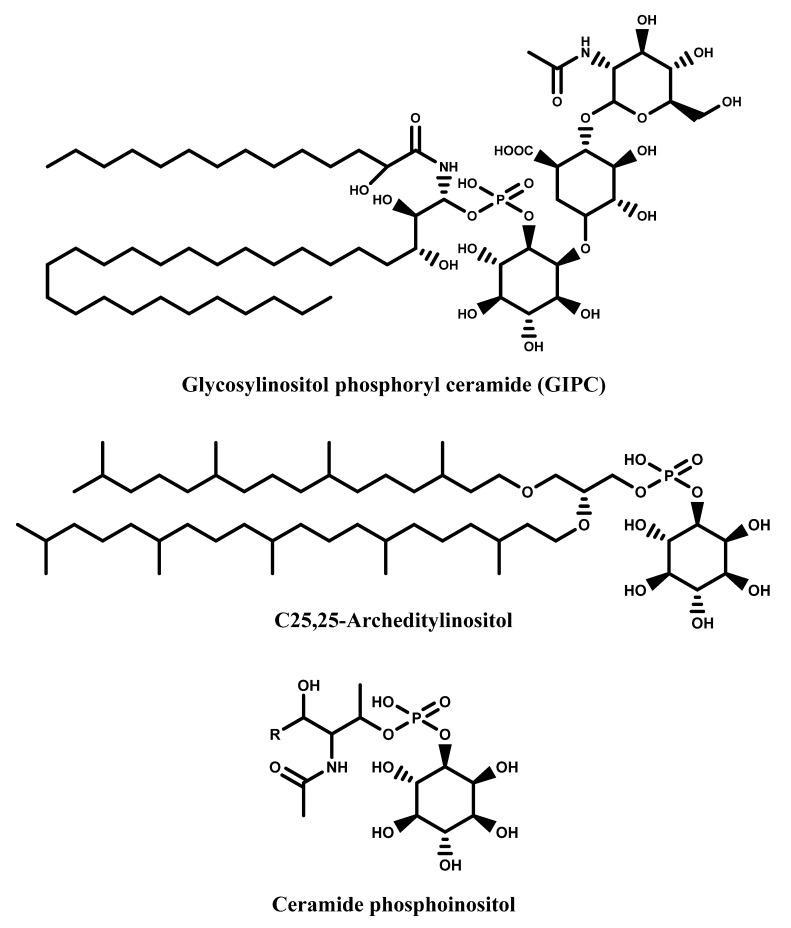
Examples of inositol ceramides, sphingolipids and archeditylinositols.

**Figure 5 molecules-27-01525-f005:**
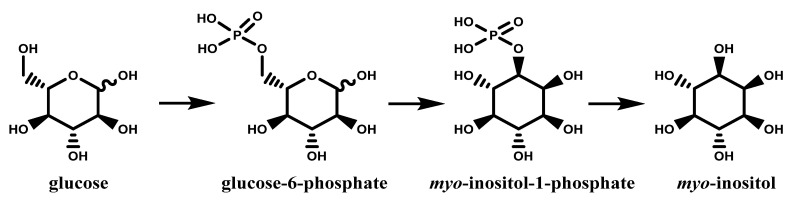
Biosynthesis of *myo*-inositol.

**Figure 6 molecules-27-01525-f006:**
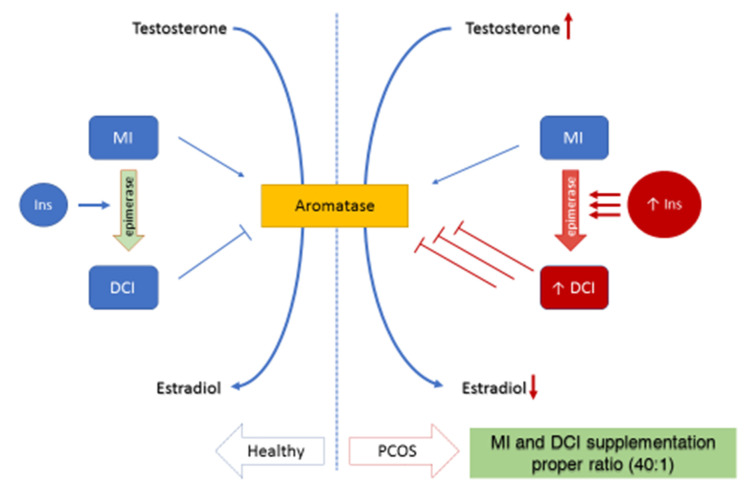
Biological functions of *myo*-inositol (MI) and *d*-*chiro*-inositol (DCI) on estrogen and androgen biosynthesis in ovarian granulosa cells of healthy and PCOS women (Ins = insulin).

**Table 1 molecules-27-01525-t001:** Occurrence of inositols in plants.

Family	Species	Inositols	Refs.
**Adoxaceae**	*Sambucus nigra*	d-pinitol, *allo*-inositol, d-*chiro*-inositol, ononitol, bornesitol, *scyllo*-inositol, *myo*-inositol	[41]
**Agaricaceae**	*Agaricus bisporus*	d-pinitol, *allo*-inositol, d-*chiro*-inositol, bornesitol, *scyllo*-inositol, *myo*-inositol	[41]
**Amarantaceae**	*Spinacia olearia*	*myo*-inositol	[42]
	*Beta vulgaris*	d-*chiro*-inositol, bornesitol, *myo*-inositol	[41,42]
**Amarylidaceae**	*Allium cepa*	d-chiro-inositol, bornesitol, *myo*-inositol	[41,42]
	*Allium sativum*	*allo*-inositol, ononitol, *myo*-inositol	[41]
	*Allium ursinum*	d-pinitol, d-*chiro*-inositol, ononitol, *scyllo*-inositol, *myo*-inositol	[41]
**Apiaceae**	*Anethum graveolens*	ononitol, bornesitol, *scyllo*-inositol, *myo*-inositol	[41]
**Apiaceae**	*Carum carvi*	d-pinitol, d-*chiro*-inositol, bornesitol, *scyllo*-inosito, *myo*-inositol	[41]
	*Daucus carota*	d-chiro-inositol, bornesitol, *scyllo*-inositol, *myo*-inositol	[41]
	*Petroselinum crispum*	d-pinitol, *allo*-inositol, d-*chiro*-inositol, bornesitol, *scyllo*-inositol, *myo*-inositol	[41]
**Apocynaceae**	*Hancornia speciosa*	Bornesitol	[50]
	*Calendula anthodium*	d-pinitol, d-*chiro*-inositol, bornesitol, *scyllo*-inositol, *myo*-inositol	[41]
**Asteraceae**	*Cichorium intybus*	d-*chiro*-inositol, *scyllo*-inositol, *myo*-inositol	[42]
	*Cichorium endivia*	d-*chiro*-inositol, *scyllo*-inositol, *myo*-inositol	[42]
	*Cichorium endivia* var. *latifolia*	d-*chiro*-inositol, *myo*-inositol	[42]
	*Cynara cardunculus*	d-*chiro*-inositol, *scyllo*-inositol, *myo*-inositol	[42]
	*Lactuca sativa*	d-*chiro*-inositol, *myo*-inositol	[42]
	*Lactuca sativa* var. *crispa*	d-*chiro*-inositol, *myo*-inositol	[42]
	*Lactuca sativa* ‘Lollo Rosso’	d-*chiro*-inositol, *myo*-inositol	[42]
	*Lactuca sativa* var. *longifolia*	d-*chiro*-inositol, *myo*-inositol	[42]
	*Lactuca sativa* ‘Red Batavian’	d-*chiro*-inositol, *myo*-inositol	[42]
	*Matricaria chamomila*	d-pinitol, *chiro*-inositol, *neo*-inositol, bornesitol, *scyllo*-inositol, myo-inositol	[48]
	*Solidago virgaurea*	d-pinitol, d-*chiro*-inositol, ononitol, bornesitol, *scyllo*-inositol, *myo*-inositol	[41]
	*Tanacetum officinale*	*myo*-inositol	[41]
	*Taraxacum officinale*	d-pinitol, d-*chiro*-inositol, bornesitol, *myo*-inositol	[41]
**Boraginaceae**	*Phacelia tanacetifolia*	*allo*-inositol, *scyllo*-inositol, *myo*-inositol	[51]
**Brassicaceae**	*Brassica oleracea*	d-pinitol, d-*chiro*-inositol, bornesitol, *myo*-inositol	[41,42]
	*Brassica oleracea* var. *sabellica*	d-pinitol, d-*chiro*-inositol, ononitol, *myo*-inositol	[41]
	*Camelina sativa*	d-pinitol, d-*chiro*-inositol, bornesitol, *myo*-inositol	[41]
	*Raphanus raphanistrum* subsp. *sativus*	myo-inositol	[42]
**Convolvulaceae**	*Ipomoea batatas*	d-pinitol, d-*chiro*-inositol, bornesitol, *scyllo*-inositol, *myo*-inositol	[41]
**Cupressaceae**	*Juniperus communis*	d-pinitol, sequoytol	[49]
**Dioscoreaceae**	*Dioscorea alata*	*scyllo*-inositol, *myo*-inositol	[42]
**Ericaceae**	*Vaccinium myrtillus*	*allo*-inositol, d-*chiro*-inositol, ononitol, bornesitol, *myo*-inositol	[41]
**Fabaceae**	*Arachis hypogea*	d-pinitol, d-*chiro*-inositol, *myo*-inositol	[41]
	*Ceratonia silique*	d-pinitol, *allo*-inositol, d-*chiro*-inositol, bornesitol, *scyllo*-inositol, *myo*-inositol	[41,43]
	*Cicer arietinum*	d-pinitol, d-*chiro*-inositol, *myo*-inositol, galactosyl-inositol, galactosyl-pinitol, ciceritol	[43,44]
	*Erythrina edulis*	d-chiro-inositol, *myo*-inositol	[44]
	*Fagopyrum esculentum*	d-*chiro*-inositol, *myo*-inositol, galactosyl-inositol, fagopyritol	[43]
	*Glycine max*	d-pinitol, d-*chiro*-inositol, *myo*-inositol, fagopyritol, galactosyl-pinitol	[43,44,54]
	*Glycyrrhiza glabra*	d-pinitol	[52]
	*Latirus sativus*	bornesitol, *myo*-inositol, galactosyl-inositol	[43]
	*Lens culinaris*	d-pinitol, d-*chiro*-inositol, bornesitol, *myo*-inositol, galactosyl-inositol, galactosyl-pinitol, ciceritol	[43,55]
	*Lupinus perennis*	d-pinitol, *allo*-inositol, d-*chiro*-inositol, bornesitol, *myo*-inositol	[41]
	*Medicago sativa*	d-pinitol, d-*chiro*-inositol, ononitol, *scyllo*-inositol, *myo*-inositol	[41,47]
	*Phaseolus polyanthus*	*myo*-inositol	[44]
	*Phaseolus vilgaris*	*myo*-inositol	[44,46]
	*Pisum sativum*	*myo*-inositol	[44]
	*Trigonella foenum-graecum*	d-pinitol, d-*chiro*-inositol, bornesitol, *myo*-inositol	[41]
	*Vicia faba*	*myo*-inositol	[44]
	*Vicia sativa*	*myo*-inostol	[44]
	*Vigna unaniculata*	ononitol, *myo*-inositol, galactosyl-inositol, galactosyl-ononitol	[43,44,45]
**Hydrophyllaceae**	*Phacella tanacetifolia*	d-pinitol, *allo*-inositol, d-*chiro*-inositol, bornesitol, *scyllo*-inositol, *myo*-inositol	[41]
**Hypericaceae**	*Hypericum perforatum*	d-*chiro*-inositol, bornesitol, *scyllo*-inositol, *myo*-inositol	[41]
**Lamiaceae**	*Mentha piperita*	d-pinitol, d-*chiro*-inositol, ononitol, bornesitol, *myo*-inositol	[41]
	*Salvia officinalis*	d-pinitol, d-*chiro*-inositol, bornesitol, *scyllo*-inositol, *myo*-inositol	[41]
**Lauraceae**	*Cinnamomum verum*	*allo*-inositol, d-*chiro*-inositol, *scyllo*-inositol, *myo*-inositol	[41]
	*Laurus nobilis*	d-pinitol, bornesitol, *scyllo*-inositol, *myo*-inositol	[41]
**Myristicaceae**	*Myristica fragrans*	d-*chiro*-inositol, *myo*-inositol	[41]
**Myrtaceae**	*Eugenia caryophyllus*	*allo*-inositol, d-*chiro*-inositol, ononitol, bornesitol, *scyllo*-inositol, *myo*-inositol	[41]
**Pinaceae**	*Abies sibirica*	d-pinitol, sequoytol	[49]
	*Larix gmelinii*	d-pinitol, sequoytol	[49]
	*Picea abies*	d-pinitol, sequoytol	[49]
	*Pinus sibirica*	d-pinitol, sequoytol	[49]
**Plumbaginaceae**	*Limonium perezii*	d-*chiro*-inositol, *myo*-inositol	[56]
	*Limonium sinvatum*	d-*chiro*-inositol, *myo*-inositol	[56]
**Poaceae**	*Oryza sativa*	d-*chiro*-inositol, *myo*-inositol	[41]
**Rosaceae**	*Rosa canina*	d-pinitol, d-*chiro*-inositol, bornesitol, *scyllo*-inositol, *myo*-inositol	[41]
	*Sorbus aucuparia*	d-pinitol, *allo*-inositol, d-*chiro*-inositol, bornesitol, *scyllo*-inositol, *myo*-inositol	[41]
**Solanaceae**	*Capsicum annuum*	*allo*-inositol, d-*chiro*-inositol, bornesitol, *scyllo*-inositol, *myo*-inositol	[41]
	*Solanum melongena*	*scyllo*-inositol, *myo*-inositol	[42]
	*Solanum tuberosum*	d-pinitol, d-*chiro*-inositol, bornesitol, *myo*-inositol	[41]
**Zingiberaceae**	*Curcuma longa*	d-pinitol, d-*chiro*-inositol, bornesitol, *myo*-inositol	[41]
	*Elettaria cardamomum*	d-pinitol, *allo*-inositol, d-*chiro*-inositol, ononitol, *scyllo*-inositol, *myo*-inositol	[41]
	*Zingiber officinalis*	d-pinitol, d-*chiro*-inositol, bornesitol, *myo*-inositol	[41]

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
