# Peer review of "Novel Chemical and Biological Insights of Inositol Derivatives in Mediterranean Plants"

_molecules, 2022, doi:10.3390/molecules27051525_

Round 1

Reviewer 1 Report

It is a well written review article which summarize the involvement of inositols in the modulation of many inflammatory processes and the insulin-like behavior of some of them makes them particularly interesting molecules from a pharmacological point of view. The authors used up-to-date literature, described results are supported by proper figures. Prepared table is informative. My minor comment is that it would be valuable to indicate the concentration of myo-inositol in analyzed medicinal and herbs.

Author Response

According to the referee’s comment we have added the concentrations of inositols in the herbs and medicinal plants from the available data reported in literature.

Sincerely

Giuseppe Ruberto

Reviewer 2 Report

The manuscript "Novel chemical and biological insights of inositol derivatives in Mediterranean plants" presents the data about inositol and its derivates, they biological source, and biological activities. A lot of structures of inositols, such as methyl ethers derivatives, glucosides, pyrophosphates, archedityl inositols, etc. are presented. 27 families of plants are considered natural sources of inositols. This information is discussed in the text and is presented in the table. Biological activities of inositols discussed in the manuscript include activity against Polycystic ovary syndrome, diabetes, neurodegenerative and neurologic disorders, cancer, and even SARS-CoV-2.

The manuscript provides sufficient information about the research of inositols over the past 5 years.

This work is interesting for fundamental science from the metabolomics point of view, and for the investigation of the source of natural bioactive compounds.

I think the manuscript may be published in the Molecules Journal in present form. 

Author Response

We would like to express our thanks for the favorable evaluation of our manuscript.

Sincerely

Giuseppe Ruberto